# Precision Medicine in Parkinson’s Disease: From Genetic Risk Signals to Personalized Therapy

**DOI:** 10.3390/brainsci12101308

**Published:** 2022-09-28

**Authors:** Giulia Straccia, Fabiana Colucci, Roberto Eleopra, Roberto Cilia

**Affiliations:** 1Neurology and Stroke Unit, Centro Traumatologico Ortopedico (C.T.O) Hospital, A.O.R.N., Ospedali dei Colli, 80131 Naples, Italy; 2Fondazione IRCCS Istituto Neurologico Carlo Besta, Department of Clinical Neurosciences, Parkinson and Movement Disorders Unit, 20133 Milan, Italy; 3Dipartimento di Neuroscienze e Riabilitazione, Università degli Studi di Ferrara, 44100 Ferrara, Italy

**Keywords:** Parkinson’s disease, precision medicine, disease-modifying therapy, genetics, GBA

## Abstract

Understanding the pathophysiology and genetic background of Parkinson’s disease (PD) increases the likelihood of developing effective disease-modifying therapeutic strategies. In particular, the discovery of genetic variants causing or increasing the risk for PD has contributed to refining the clinical, biological, and molecular classification of the disease and has offered new insights into sporadic forms. It is even more evident that specific genetic mutations can show different responses to pharmacological and device-aided therapies. To date, several agents acting on multiple PD-causing pathogenic pathways have been tested as disease-modifying strategies, with disappointing results. This may be caused by the recruitment of PD populations whose underlying molecular pathophysiology is heterogeneous. We believe that an effective model of personalized medicine must be prioritized in the near future. Here, we review the current therapeutic options under clinical and preclinical development for PD and discuss the key pending questions and challenges to face for successful clinical trials. Furthermore, we provide some insights into the role of genetics in guiding the decision-making process on symptomatic and device-aided therapies for PD in daily clinical practice.

## 1. Introduction 

Parkinson’s Disease (PD) currently displays the fastest growing prevalence among neurodegenerative diseases, as its prevalence doubled from 1990 to 2015 [1]. It has been estimated that—if PD prevalence maintains this growth rate—nearly 13 million people will be affected by 2040 [1]. Due to the actual epidemiological burden and the expected increase in average lifespan in the future, PD exerts a massive impact on both social and individual levels, being a significant source of disability, poor quality of life, and socioeconomic cost [1]. PD has been initially considered as a pure disorder of movement due to the progressive degeneration of nigrostriatal dopaminergic neurons in the substantia nigra pars compacta (SNpc), secondary to the accumulation of insoluble α-synuclein (α-Syn) into abnormal neuronal inclusions called Lewy Bodies [2,3,4]. Nevertheless, the combination of motor (e.g., bradykinesia, rigidity, tremor, postural instability) and non-motor symptoms (e.g., hyposmia, autonomic dysfunction, sleep disorders, constipation, depression, cognitive decline) [5] clarified that PD is a multisystem disorder with complex neuropathology, affecting both the central and peripheral nervous system. 

Despite there still being uncertainties, substantial progress has been made in understanding PD pathogenesis, starting from identifying gene–environmental interactions—influencing the clinical severity and individual age-related predisposition of developing PD—to specific molecular pathways, promoting dysfunction at a cellular/sub-cellular level and conditioning selective neuronal vulnerability. These findings support the concept that PD is not a unique and stereotyped clinical–biological entity but a heterogeneous multifactorial disease spectrum. Applied to disease-modifying strategies, this notion opens the way to multiple differential approaches ideally targeting different pathophysiological mechanisms or predisposing factors, but also imposes special requirement to refine the stratification of patients, to identify reliable biomarkers reflecting early neurodegeneration, disease severity and progression, engagement of targets and therapy outcomes [6]. 

In this scenario, it is crucial to establish whether certain dysfunctional pathways are truly pathogenic or derive from compensatory or neuroprotective efforts of the brain [6]. Considering the multiple failures of all clinical trials on disease-modifying therapies in PD over the last 20 years, we are currently moving towards an era of precision medicine, which is not only “personalized”, namely aimed to tailor treatment to a specific individual and group characteristics, but also “precise”, as grounded in a more refined clinical, genetic, biological and molecular definition of the disease. Oncology offers a successful example of this approach, as therapies are primarily based on tumor grading, histology, molecular and hormonal profile, and mechanisms of tumor resistance to treatment are specifically investigated [7]. Advances in therapy should proceed in parallel with advances in disease knowledge. 

Recent advances into the genetic underpinnings of PD have proven extremely useful in promoting a more precise diagnosis, as they provide an “etiopathogenetic signature” of the disease. Moreover, the heterogeneity of the genetic landscape in PD confirms the complexity of its molecular and biological pathways and offers new insights into sporadic forms. Indeed, whereas genetic-driven current clinical trials only target PD patients carrying mutations in specific genes, some of the therapies could also be promising in sporadic patients due to common etiopathogenetic mechanisms.

## 2. The Genetic Architecture of PD 

The discovery of the first disease-causing missense mutation in the SNCA, a gene encoding for alpha-synuclein (α-Syn), in a large Italian kindred and three unrelated families of Greek ancestry [8], opened the way to identify several other genes linked to autosomal dominant and autosomal recessive familial PD [9]. To date, approximately 20 genes have been associated with PD [9]. However, only some of these genes (e.g., SNCA, LRRK2, VPS35, PRKN, PINK1, GBA, and DJ-1) have been convincingly demonstrated as being related to PD pathogenesis, while the role of other genes is still debated and validation is still required (e.g., LRP10 [10,11], TMEM230 [12,13,14], DNAJC13, UCHL1, HTRA2, GIGYF2, and EIF4G1 [15]). 

Genome-wide association studies (GWAS) and extensive sequencing approaches have further expanded the genetic background of PD [9]. The most extensive meta-GWAS study to date, performed on 7.8 million single nucleotide polymorphisms (SNPs), 37,700 PD cases, 18,600 UK Biobank healthy individuals with positive PD family history, and 1.4 million controls, identified 90 independent risk signals across 78 genomic regions, collectively explaining 16–36% of the genetic PD risk [16]. Interestingly, most of the loci were located close to genes involved in monogenic PD. In contrast, others contained putative causal genes involved in brain innate immunity (GRN) [17], endocytic pathways (VAMP4 and NOD2) [18], lysosomal storage (GRN, GUSB, and NEU1) [17], pathways that have been linked to the pathogenesis of sporadic PD [18,19,20]. Overall, GWAS loci, along with rare and common coding variants, contribute to a cumulative genetic risk of 25% of developing PD [9,21]. These genes could be classified according to their specific Mendelian inheritance pattern and penetrance rate. Overall, the monogenic causes of PD are rare, accounting for <1% of PD cases, even for the most common genes [22]. 

## 3. Autosomal Dominant Genes with High Penetrance

### 3.1. SNCA (Alpha-Synuclein; OMIM*163890)

The SNCA gene (Chr. 4p22.1) encodes for α-Synuclein (α-Syn), a 140-amino-acid highly conserved protein abundantly expressed in neurons, mainly in presynaptic terminals. Its functions remain poorly understood, but shreds of evidence support its role in recycling vesicles, synaptic plasticity, and neurotransmitter release through membrane interaction [23]. In addition, α-Syn play a role in the immune system. Specific α-Syn peptides are recognized by T lymphocytes [24] (Figure 1) and T-lymphocytes induced by α-Syn epitopes are overexpressed in PD compared to controls, suggesting a potential role of α-Syn in driving pro-inflammatory responses [24]. Although the native state of α-Syn is under debate, studies suggest that α-Syn exhibits conformational plasticity in a dynamic equilibrium among factors accelerating or inhibiting the misfolding and aggregation [23,25] into Lewy Bodies (LBs), which are considered the pathological hallmarks of PD [2]. Furthermore, misfolded synuclein may propagate through a prion-like mechanism that eventually aggregates and spreads neurodegeneration [23]. Specifically, some point mutations in the SNCA gene and several post-translational modifications, including tyrosine nitration, ubiquitylation, C-terminal truncation, and phosphorylation of serine 129 (pS129), can promote α-Syn fibrillation tendency [25,26,27]. The pS129 is the dominant α-Syn form in LBs [28], as it has been detected in the brains of PD patients and in in vivo and in vitro models of PD [25,26], and its reduction decreases α-Syn pathology in in vitro experiments [29]. Similarly, α-Syn nitration at residues Y39, Y125, and Y133 favors the accumulation/precipitation of α-Syn and stabilizes oligomers, potentially enhancing their toxicity, inducing degeneration of dopaminergic neurons in animal models. This α-Syn form has been revealed in patients with synucleinopathies [27]. The truncated forms of α-Syn have also been found in LBs [30]. Truncation typically concerns the C-terminal portion of α-Syn, and it is associated with a high propensity of α-Syn to fibrillation in vitro [27,31]. Immunotherapies against truncated α-Syn forms and inhibition of caspases reduce α-Syn pathology and favor neuroprotection in transgenic animal models of synucleinopathies [32,33]. Nevertheless, not all of the truncated forms of α-Syn are toxic, and truncated forms are found in healthy brains as well [34]. 

The first SNCA gene mutation (p.A53T, OMIM*163890.0001) responsible for autosomal dominant (AD) familial PD was identified in a large kindred with Italian and Greek ancestry (Contursi kindred) [8]. Since then, other SNCA punctiform mutations, such as p.A30P, p.E46K, p.H50Q, p.G51D, p.A53E [36,37,38,39,40,41], and SNCA locus rearrangements responsible for gene duplications and triplications [42,43,44], have been associated with PD or Lewy body dementia (LBD). The overall SNCA mutations have been described only in a few families. The clinical spectrum changes according to mutation type, ranging from forms resembling idiopathic PD to more severe phenotypes. The latter generally is observed in gene duplication and triplication carriers, with mainly an early age of onset, rapid motor progression, and high non-motor symptoms burden (dysautonomia, hallucinations, dementia) [45,46,47,48,49]. Moreover, a large GWAS study reported that only specific SNCA transcripts correlate with the risk of developing dementia in synucleinopathies [50]. The central role of α-Syn in PD pathogenesis is further supported by the recent evidence that non-pathogenic SNCA gene polymorphisms influence motor phenotype and age at onset in other monogenic forms of PD [51,52], and polymorphisms in non-coding regions adjacent to SNCA gene, potentially affecting gene expression and protein translation, increase PD risk [53]. These clinical data support the causative role of wild-type α-Syn overexpression and suggest a dose-dependent correlation between α-Syn levels and clinical severity [30,54].

#### α-Synuclein as Therapeutic Target

α-Syn therapeutic modulation aims to either reduce and/or prevent the formation of toxic α-Syn aggregates, meaning by α-Syn synthesis and aggregation inhibition, degradation enhancement, reduction in α-Syn membrane displacement, and α-Syn-directed active and passive immunization (Table 1). 


**(a) α-Synuclein synthesis inhibitors**


The therapeutic strategies for reduction in total α-Syn amount (synthesis inhibitors) comprise silencing mechanisms aimed at reducing α-Syn intracellular mRNA synthesis [55], such as Antisense Oligonucleotides (ASOs), short interfering RNAs (siRNAs), and histone acetylase modulators. The recent reports of successful phase III trials using ASO in spinal muscular atrophy [121] and Huntington’s Disease [122] have encouraged these therapeutic approaches. ASOs against SNCA reduce SNCA mRNA levels, decrease deposition and spread of phosphorylated α-Syn, and loss of Tyrosine Hydrolase activity in rodent animal models based on intrastriatal injection of α-Syn fibrillar fibrils [55]. Notably, human SNCA-targeting ASOs display broad target engagement in non-human primate brains and decrease α-Syn levels in the cerebral spinal fluid (CSF), providing new hope for future assessment in humans [55]. 

Regarding the use of viral vector-mediated siRNAs against α-Syn, adeno-associated virus (AAV)-mediated delivery of short hairpin RNA (shRNA) inhibits the expression of endogenous α-Syn in SNpc, attenuating the motor deficit of rotenone-exposed and decreasing the rotenone-induced degeneration of nigral dopaminergic neurons in adult rats [88,89]. 

A new amido-bridged nucleic acid (AmNA)-modified ASO has been recently developed that downregulates SNCA mRNA and protein levels in in vitro and in vivo PD models finally leading to improved neurological deficits [65]. Similarly, an exosome-ASO (Exo-ASO4) was able to reduce α-Syn expression and aggregation as well as dopaminergic neurons degeneration in an α-Syn A53T mice model, resulting in improved motor functions [84]. Studies on animal models suggest that both up- and downregulation of endogenous α-Syn expression could contribute to nigrostriatal degeneration, potentially due to impairment of physiological α-Syn functions [89]. Further studies are necessary to clarify the extent to which the toxicity of ASOs or siRNAs could be attributed to an excessive reduction in endogenous α-Syn levels and how silencing strategies could be well balanced. 

Another interesting therapeutic strategy to reduce α-Syn expression is the administration of 2-adrenergic receptor (2AR) agonists: in mice models of neurotoxin-induced parkinsonism and induced pluripotent stem cells (iPSC)-derived neuronal cultures, β2AR-agonists reduced SNCA expression, modulating gene expression through histone acetylation, and mitochondrial free radicals [84]. The effect of β2AR modulation on PD risk has raised interest [94,95,96,123] because of the reduced vs. increased risk of developing PD in individuals using salbutamol (β2AR agonist) vs. propranolol (β2AR-antagonist), respectively [94,95]. It has been suggested that the use of β2AR agonists in PD patients has been evaluated only in small open-label studies, detecting a reduction in Parkinsonian symptoms and daily levodopa dose [124], increased duration of daily total ON [125], and response to Levodopa [126]. Although large, randomized, controlled trials are still needed to clarify the effects of β2AR agonists on the risk of PD [127], it is worth highlighting that H3K27 acetylation of SNCA can be regulated in a level-dependent manner by both β2AR agonists (down) and antagonists (up) [94].

Finally, the SNCA synthesis might be reduced by directly targeting the SNCA mRNA through epigenetic modulators. Small molecules inhibit SNCA cellular translation by targeting iron-responsive elements located in the 5′ untranslated regions (5′ UTR) of the SNCA gene. The most potent small molecule under study is Synucleozid, which decreases polysomes’ SNCA mRNA load and exerts a cytoprotective effect in vitro [109]. Other small molecules targeting histone acetylation and deacetylation could act as pharmacological modifiers of the epigenetic state. For example, histone deacetylation suppresses transcriptional activity and its inhibition seems to deteriorate DA neuronal function and upregulates SNCA expression [105]. A phase I trial of the FDA-approved histone deacetylation inhibitor, Glycerol Phenylbutyrate, preliminarily reported increased plasma concentrations of a-synuclein in both the PD patients and healthy age-matched controls (ClinicalTrial.gov ID: NCT02046434), suggesting that this mechanism of action may increase the clearance of α-syn from the human brain.


**(b) Inhibiting α-synuclein aggregation**


Therapeutic approaches aimed at reducing or inhibiting α-Syn aggregation are based on the assumption that aggregates are neurotoxic. Nevertheless, it is still debated if aggregates are causative or compensatory [6]. Despite these controversies, several compounds have been studied for their ability to prevent α-Syn aggregation. 

Intrabodies, antibody fragments (140–250 amino acids) binding to intracellular proteins, are gaining significant scientific interest in neurodegenerative proteinopathies [128]. Intrabodies can be delivered both as genes and proteins and engineered to target monomeric α-Syn [128]. In rodent models with a viral-vector-mediated α-Syn overexpression, intrabodies reduced α-Syn aggregates and nigrostriatal neurodegeneration [56]. Notably, two proteasome-directed intrabodies, VH14*PEST (targeting α-Syn non-amyloid component region) and NbSyn87*PEST (directed against α-Syn C-terminal region), injected into SN of overexpression-based PD rodent models, markedly reduced the level of phosphorylated pS129 α-Syn, increased TH immunoreactivity, dopamine transporter density, and improved motor functions [57]. 

Different compounds able to inhibit α-synuclein aggregation are small molecules [66]. In vitro studies showed that Rifampicin and its derivatives inhibited the aggregation of α-synuclein [67] and Aβ [68]. Baicalein is able to bind α-syn and inhibit its nucleation, suggesting a possible role as a stabilizer of semi-folded α-synuclein [69]. The stabilization in vitro seems to be a valid therapeutic approach [70,71,72], as well as the conversion of mature α-synuclein fibrils into smaller non-toxic forms [73]. Further promising data, currently in phase I trials, come from other biologic small molecules that inhibit α-Syn misfolding and aggregation (Table 1), including NPT200-11 [90,91], Anle138b [99,129,130,131,132], Leuco-methylthioninium bis [106], CLR01 [110], KYP-2047 [112,113,114,115], NPT100-18A [117], cyclized nordihydroguaiaretic acid [118], Fasudil [119], and Squalamine [120]. Finally, heat shock proteins have been identified as components of LBs and some of them are able to suppress the formation of α-Syn fibrils in vitro [133]. 

Although several of the abovementioned strategies based on small molecules might be promising disease-modifying compounds inhibiting α-Syn aggregation, clinical trials in humans are still lacking.


**(c) Inhibiting α-Synuclein uptake**


The propagation in a prion-like manner of pathological α-Syn aggregates through synaptically and anatomically connected brain regions, due to exocytosis and uptake of α-Syn aggregates from the extracellular space by nearby neurons, has been proposed as a mechanism favoring proteinopathy progression [23,85]. The proper mechanism by which α-Syn-aggregated extracellular proteins bind and enter close cells to trigger intracellular fibril formation is still unclear, despite the active clathrin endocytic process that has been suggested as playing a role [74]. 

Previous studies on prion proteins have suggested that the transcellular transmission of prion aggregates is facilitated by the binding to heparan sulfate proteoglycans on the cell surface [134,135]. Using recombinant preformed fibrils of α-Syn to investigate extracellular protein-binding partners promoting α-Syn transcellular propagation, three candidates have been identified, such as membrane proteins binding lymphocyte-activation gene 3 (LAG3), neurexin 1β, and amyloid-beta precursor-like protein 1 [74]. Among them, LAG3 exhibited the highest and most specific affinity for α-Syn preformed fibrils over other monomers and misfolded aggregates. Interestingly, in cellular and animal models, deletion of LAG3 or administration of LAG3-directed antibodies inhibited the internalization of α-Syn preformed fibrils, preventing transcellular α-Syn propagation and neurotoxicity [74]. 


**(d) Promoting clearance of α-Synuclein: Immunotherapies and Autophagy-enhancing agents**


-
*
Immunotherapies
*


Due to the preferential action of antibodies on extracellular proteins, immunotherapy is a promising strategy to promote immune-mediated degradation of extracellular α-Syn aiming to reduce its spreading. Clinical trials currently use two types of α-Syn immunotherapy: passive and active immunization. Several compounds have been investigated (Table 1), with disappointing results so far. The former is obtained by administering antibodies specific to α-Syn, whereas the latter is based on the production of endogenous antibodies following the injection of modified α-Syn (e.g., vaccination). Due to a higher potential risk of adverse effects by active immunization, more advances have been achieved in passive immunotherapy. 

Passive immunotherapy. Several preclinical studies have demonstrated that α-Syn-directed antibodies (more than 50 have been developed) decrease α-Syn aggregation in α-Syn-overexpressing mice and prevent behavioral deficits. Clinical trials in humans investigated the efficacy and safety of Cinpanemab/BIIB054 and Prasinezumab/RO7046015/PRX002. Cinpanemab is a fully human IgG1 monoclonal antibody with high selectivity for aggregated forms of α-Syn (almost 800-fold higher affinity for fibrillary versus monomeric α-Syn forms); however, the phase 2 trial on Cinpanemab (SPARK study) failed to meet primary and secondary outcome measures [106,107,108]. Similarly, the first phase 2 RCT on Prasinezumab (PASADENA Study) failed to meet the primary endpoint [75]. A phase 2b trial on intravenous Prasinezumab in early PD (PADOVA study) is ongoing (ClinicalTrials.gov ID: NCT04777331). 

Active immunotherapy. Affitope PD01A/PD03A is the compound in the most advanced study phase. PD01A is an 8-aminoacids antigenic peptide that, mimicking the C-terminal region of human α-Syn, promotes endogenous active immune response and the synthesis of antibodies directed against α-Syn aggregates, with much lower affinity to α-Syn monomeric forms [107]. In a phase 1, patient-blinded, pilot study, repeated doses of PD01A showed suitable active immunization in 24 PD patients (of which 21 patients received all 6 PD01A doses), with minor side effects and good immunogenicity [107]. PD01A-induced antibodies preferentially targeted both oligomeric and fibrillary α-Syn forms and reduced CSF and plasma oligomeric forms in patients receiving high-dose immunization [107]. Nevertheless, no changes in clinical scores were detected [107]. A phase 1 study on PD01A on PD carriers of mutations in the Glucocerebrosidase gene (GBA), was withdrawn in October 2017, before recruiting the first participant (ClinicalTrials.gov ID: NCT02758730). 

A parallel phase 1 study has been conducted on high dose PD01A and PD03A in 30 patients with early MSA (ClinicalTrials.gov ID: NCT02270489). The majority of patients reported adverse events, the most frequent related to site reactions [128]. Notably, PD01A induced more frequently an immune response and showed higher immunogenicity compared to PD03A [128]. 

-
*
Autophagy-enhancing agents
*


Autophagy is a complex cellular pathway that intervenes in the degradation of misfolded proteins and damaged organelles, contributing to cellular homeostasis [136]. Several shreds of evidence suggest that autophagy is involved in the clearance of intracellular α-Syn aggregates, along with the ubiquitin-proteasome system (UPS) [137]. Autophagy encompasses several distinct pathways (including macroautophagy, chaperone-mediated autophagy, and microautophagy) that need delivery of intracellular constituents into lysosomes and the subsequent result blocks reutilization. Therefore, a bidirectional loop has been proposed for α-Syn-induced toxicity: the pathogenic α-Syn species accumulate as a consequence of autophagy dysfunction, which in turns favors further autophagy impairment due to abnormal α-Syn reutilization [137]. In this scenario, enhancing autophagy/lysosome pathways could interrupt this vicious cycle, protecting against neurodegeneration [137]. Several molecules have been tested (Table 1), including Rapamycin [62,63,64], and Nilotinib [76]. 

Other minor compounds targeting autophagy comprise mitochondrial pyruvate carriers (MPC) inhibitors, which are in very early development [100,101,102,103,104]. Apart from autophagy, the UPS system contributes to α-Syn degradation [62]. As for autophagy, bidirectional crosstalk is hypothesized between α-Syn and UPS, as misfolded α-Syn accumulation directly impairs normal UPS functioning [138]. Increasing UPS activity could have a therapeutic potential to promote α-Syn clearance; indeed, inhibition of UPS14, a proteasome-associated deubiquitinase, through small-molecules UPS enhancers, has been demonstrated to accelerate the degradation of oxidized proteins and increase resistance to oxidative stress [139,140].

## 4. Autosomal Dominant Genes with Variable Penetrance

### 4.1. LRRK2 (Leucine-Rich Repeat Kinase 2; OMIM*609007)

The leucine-rich repeat kinase 2 (LRRK2) gene encodes for a large (2527 amino acids) multidomain protein widely expressed in the brain, heart, kidney, lungs, and detectable in several biofluids (urine, CSF, blood). LRRK2 has two major enzymatic properties, a GTPase function due to a Ras-of-Complex (Roc) GTPase domain and a tyrosine–kinase function linked to the C-terminal-of-Roc domain of the protein [141]. LRRK2 mutations are the most common genetic variants that are the cause of familial PD. Almost 100 variants in LRRK2 have been identified, with only a few of them associated with PD by linkage studies [142]. Mutations induce neuronal damage in culture models, and rodent LRRK2 mutation models show degeneration of SNpc dopaminergic neurons [141]. Mutations linked to PD generally cluster within GTPase (e.g., p.Y1699C, p.R1628P, p.R1441C/G/H, and p.N1437H) or kinase enzymatic domains (e.g., p.G2019S, p.I2020T, and p.I2012T), confirming their impact on protein functions [141]. All of these variants are very rare, except for p.G2019S (OMIM*609007.0006), which accounts for almost 1% of sporadic PD cases and 4% of familial PD cases worldwide [143]. In specific populations the prevalence significantly increases, reaching 29% and 37% of familial PD cases in Ashkenazi Jewish and North Africans, respectively [144]. Some pathogenic mutations are further specific to selected populations, such as p.R1441G (OMIM*609007.0003) which is responsible for nearly 46% of familial Basque ascent PD cases, due to a founder effect [144,145]. LRRK2 mutations display variable penetrance, depending on age, ancestral genetic background, and environment [144]. Indeed, p.G2019S mutation penetrance is estimated to vary from 20% in Norwegians to 60% in Tunisians by the age of 60 years [146], from 42.5 to 74% by the age of 80 years in non-Ashkenazi Jewish populations [147], and 25% in Ashkenazi Jewish populations [148]. Other LRRK2 mutations increase the risk of developing PD with variable penetrance [149]. 

Clinically, LRRK2-related PD strongly resembles idiopathic PD [144], despite some reports suggesting a slower motor progression in p.G2019S mutation carriers [143,145,150] and an overall, less severe burden of non-motor symptoms compared to PD noncarriers [144,151]. 

LRRK2 dysfunction induces abnormal kinase activities, influencing the accumulation of α-syn and its pathology to alter cellular functions and signaling pathways [141,152]. The exact mechanism by which increased kinase activity can induce PD is not known, but impairment in several cellular processes is advocated, including autophagy, lysosomal function, mitochondrial function, vesicular transport, neurotransmission, and neuroinflammation (Figure 2) [141]. 

#### Targeting LRRK2 in Clinical Trials

In transgenic mice, the p.G2019S mutation induces over-phosphorylation of mitogen-activated protein kinase (MKK)-4 and extracellular signal-regulated kinases (ERK), leading to a degeneration of SNc dopaminergic neurons and the downregulated transcription of α-synuclein, respectively [152]. In cellular models, the p.G2019S mutation exhibits a gain-of-function, enhancing LRRK2 kinase activity [152] and leading to mitochondrial DNA (mtDNA) damage, which was not observed with mutations not affecting kinase activity (e.g., p.D1994A) [153]. Indeed, in a LRRK2 p.G2019 mutation mouse model, treatment with the LRRK2 kinase inhibitor induced reduction in TH-positive dopaminergic neurons in a kinase-dependent manner (through GW5074 and indirubin-3′-monooxime) [154], and prevented or restored mtDNA damage (through GNE-7915) [153]. 

Similarly, in AAV-induced LRRK2 rat models, the expression of the p.G2019S mutation selectively induced neuronal misfolded protein inclusions and neurite degeneration, while the introduction of a kinase-inactive mutation ameliorated striatal neuropathology [155], as well as the administration of the selective kinase inhibitor PF-360 and modulation of GTPase activity-attenuated degeneration of SN and dopaminergic neurons [141]. The R144G LRRK2 mutation compromises cell viability, altering the autophagy mechanism and leading to mitochondrial and endoplasmic reticulum stress [156].

Notably, the impairment of LRRK2 kinase activity has been demonstrated not only in PD carriers of LRKK2 mutations but also in sporadic PD noncarriers [157], and in other genetic forms. Indeed, PD patients carrying mutations in vacuolar protein sorting-associated protein 35 (VPS35) display increased phosphorylation in Rab10, a substrate of LRRK2 [158]. Furthermore, the inhibition of LRRK2 kinase activity restored mitochondrial dysfunction induced by PARK2 and PINK1 mutations in in vivo cellular models [159], and reduced α-Syn trafficking, with an improvement of α-Syn pathology in LRRK2 noncarriers in in vivo animal models [160], 

These preclinical data suggest that kinase inhibition and modulation of GTPase activity could be promising therapeutic targets in both LRRK2-PD and other PD forms. 

Several compounds targeting LRRK2 kinase activity had been developed, with progressively more selective and powerful properties [161,162,163,164,165,166,167]. MLi-2 and PFE-360 displayed a high selectivity and potency as LRRK2 kinase inhibitors with an excellent BBB penetration [168,169]. In animal models, Mli-2 was well tolerated and demonstrated high selectivity and dose-dependent target engagement, reducing LRRK2 kinase activity by 90% [168]. Nevertheless, it did not restore pathological changes induced by mitochondrial dysfunction in a PD mouse model [168]. Similar results were obtained with PFE-360, which determined almost complete inhibition of LRRK2 kinase activity and restored a normal firing pattern in the subthalamic nucleus of rats with AAV-induced α-Syn overexpression [169,170]. Nevertheless, effects on motor outcome measures are still inconsistent and further data are needed [171]. 

Currently, two LRRK2 kinase inhibitors are in advanced therapeutic development. 

The first compound DNL201 was studied both in healthy individuals (ClinicalTrials.gov ID: NCT04551534) and in PD with or without LRRK2 mutations (ClinicalTrials.gov ID: NCT03710707). The data on healthy subjects are still not available (study concluded in August 2018). In PD individuals, this compound was well tolerated and demonstrated good target and downstream pathway engagement, including effects on lysosomal biomarkers [172]. A similar compound, DNL151, is currently under study. The safety, tolerability, pharmacokinetics, and pharmacodynamics of DNL151 have been investigated in PD patients in a phase 1b, multicenter, randomized, placebo-controlled, double-blind study (ClinicalTrials.gov ID: NCT04056689), whose results are yet to be published.

Despite promising results, some challenges remain regarding the clinical use of LRRK2 kinase inhibitors. First of all, there are safety concerns coming from preclinical studies, due to evidence of morphological changes induced by LRRK2 kinase inhibitors on pneumocytes and kidneys in animal models, although reversible at drug withdrawal [168,169,173]. Nevertheless, concerns remain for therapeutic use in PD, due to the necessity of long-term treatment. 

A potential way to overcome this limitation could be LRRK2 inhibition through antisense oligonucleotides (ASO) and nanoantibodies, that bind to different LRRK2 domains. To date, the data obtained from preclinical studies look promising, [161,174,175], whereas the data on PD patients (with and without LRRK2 mutations) are yet to come. A phase 1 study is ongoing on the ASO-BIIB094, administered intrathecally (ClinicalTrials.gov ID: NCT03976349).

Another relevant challenge is the lack of specific biomarkers estimating LRRK2 kinase activation levels and therapeutic target engagement [176]. Acquisition of these markers is of primary importance to improve patient selection, balance the correct level of kinase inhibition to determine benefits without toxic effects, and correlate kinase inhibition to clinical outcomes. 

### 4.2. GBA (Beta-Glucocerebrosidase Acid; OMIM*606463)

The GBA gene encodes for the 497-amino acid lysosomal hydrolase beta-glucocerebrosidase (GCase) responsible for hydrolyzation of glucosylceramide into glucose and ceramide and the cleavage of glucosylsphingosine and other beta-glucosides [177]. Homozygous and compound heterozygous GBA mutations, which impair enzymatic GCase function, cause the autosomal recessive lysosomal disorder Gaucher’s Disease (GD) [178]. According to the severity of the disease and the presence of neurological involvement, GD is traditionally classified into three clinical subtypes. Type 1 GD is defined as non-neuronopathic GD, lacking central nervous system manifestations, whereas GD type 2 and type 3 are classified as acute and chronic neuronopathic forms, based on the rapid or slowly progressive course of neurologic symptoms, respectively [178]. More than 500 mutations and gene rearrangements in the GBA gene have been reported to date [179], (http://www.hgmd.cf.ac.uk/ac/index.php; access date 25 April 2022), currently classified according to their biochemical effect on enzymatic activity: mild mutations (e.g., p.N370S, p.R496H), which are associated with non-neuronopathic GD in homozygous carriers, and severe mutations (e.g., p.84GG, IVS2_1, p.V394L, p.D409H, p.L444P, RecTL), which are associated with neuronopathic GD in homozygous carriers. On the other hand, there are GBA risk variants, which do not cause GD in the homozygous state but increase PD risk nonetheless; complex mutations, namely two or more variants in cis resulting from complex gene rearrangements; and unknown variants [180,181]. 

Heterozygous GBA mutations are currently recognized as the most frequent genetic risk factor for PD [182,183], with a three-fold higher risk of developing PD dementia (PDD) and Dementia with Lewy Bodies (DLB) [184,185,186,187]. It is estimated that 5–10% of PD patients are carriers of GBA mutations worldwide (Odds Ratio: 3–4.7 for mild mutations, 14.6–19.3 for severe mutations [187]) [182,184], with even higher prevalence rates in specific ancestries [188]. Four missense GBA variations (p.E326K, p.T369M, p.N370S, and p.L444P) account for almost 82% of all mutant alleles found in PD [189].

Genotype/phenotype correlation places along a continuum of severity, according to GBA mutations’ differential effect on residual GCase enzymatic activity. Nevertheless, significant intra- and inter-individual phenotypic variability makes precise genotype–phenotype correlation difficult [177]. Globally, severe GBA mutations cause a more aggressive clinical phenotype, characterized by an earlier age at onset, more rapid axial motor progression, and more serious non-motor symptoms (hyposmia, sleep disturbances, dysautonomia, hallucinations, cognitive decline, and dementia) [177,181,190], consistent with the pathological, biochemical, and imaging biomarkers, showing more prominent nigrostriatal degeneration, reduction in neocortical metabolism [190], and lower levels of cerebrospinal fluid α-Syn [191]. 

The mechanisms that link GBA mutations to PD have not been fully elucidated, but a complex and bidirectional interplay between GCase and α-Syn has been proposed [177]. Abnormal GCase activity impairs lipid homeostasis and membrane stability, alters endoplasmic reticulum function, increases oxidative stress, and further reduces proper GCase maturation, and impairs lysosomal, autophagy, and UPS functions [177,192]. These alterations result in enhanced α-Syn seeding into toxic insoluble species and promote α-Syn aggregation and cell-to-cell transmission [192]. On the other hand, α-Syn aggregates co-localize with GCase, further inhibiting proper GCase translation to endoplasmic reticulum and lysosomes, a critical step for proper maturation and function [177]. Interestingly, a significant reduction in GCase activity has also been demonstrated in PD individuals who do not carry any GBA mutation [193,194], with residual GCase activity levels correlating with clinical phenotype severity and disease progression [195]. In addition, other genetic PD pathways control GCase activity: LRRK2 mutations reduced GCase activity in cellular models. Indeed, in GBA mutant cells, the LRRK2 kinase inhibition modulates GCase activity [196], and the co-presence of GBA and LRRK2 mutations impact on age at onset and clinical phenotype [196]. 

#### 4.2.1. Substrate Reduction Therapy

Enzyme replacement therapy is efficiently used to treat systemic manifestations of GD, dramatically improving clinical outcome [197]. However, enzyme replacement therapy is not useful for the treatment of neurologic manifestations associated with GBA mutations because they do not cross the BBB. To overcome this issue and reduce pathological glycosphingolipids’ accumulation in the brain, an effective strategy might include the direct inhibition of glucosylceramide synthetase enzyme leading to substrate reduction. In a mouse model of neuronopathic GD, systemic administration of the Glucosylceramide synthetase inhibitor GZ667161 reduced brain glucosylceramide levels, improved brain pathology, and survival [198]. Similarly, in mouse models of synucleinopathies associated or not with GBA mutations, GZ667161 demonstrated good brain penetration and target engagement, significantly reduced α-Syn pathology, and improved cognitive deficits [199,200]. 

Based on these preclinical evidences, a phase 2, multicenter, randomized, double-blind, placebo-controlled trial on Venglustat (GZ/SAR402671) was carried out in early PD patients who carried GBA mutations (ClinicalTrials.gov ID: NCT02906020; MOVES-PD Trial). Venglustat showed a favorable safety and tolerability profile, reached a good target engagement in CSF determining a dose-dependent reduction in CSF glucosylceramide levels [201]. Despite promising pharmacokinetic and pharmacodynamic results, the study failed to demonstrate any symptomatic or disease-modifying effect of this compound [201].

#### 4.2.2. Increasing Glucocerebrosidase Enzymatic Activity

Given that GBA mutations induced a reduction in GCase enzymatic activity with subsequent abnormal substrate accumulation [199], restoring normal GCase enzymatic function would be a promising therapeutic strategy. At a preclinical level, increasing GCase levels in the brain ameliorated or even reversed pathological and behavioral abnormalities, induced by decreasing GCase activity [199,202], providing a strong rationale supporting the beneficial role of GCase augmentation strategies. This increase could be achieved by facilitating GCase transport into the lysosomes or correcting GBA gene defects through gene therapy. 

A first therapeutic GCase augmentation facilitating its transport into the lysosomes might be prompted by brain-penetrant small molecule chaperones, which may be classified as inhibitory, non-inhibitory, and mixed-type. *N*-(n-nonyl) deoxynojirimycin—a chaperone belonging to the iminosugars family—is the first of the inhibitory small molecule chaperones described. In GBA-mutated fibroblast cell cultures, this compound increased GCase activity, promoting GCase transit to the endoplasmic reticulum, and entrance into lysosomes [203]. Other iminosugars have been evaluated, showing the ability to increase GCase activity and restore mitochondrial dysfunction in preclinical models [204,205]. A similar response was obtained by isofagomine (afegostat-tartrate, AT2101), which increased GCase activity in N370S fibroblasts by several mechanisms (including GCase transport to the endoplasmic reticulum, GCase active lysosomal pool augmentation, and increase in its catalytic properties) [206], in GD patient-derived lymphoblastoid cell lines, and in L444P-mutated fibroblasts, by promoting GCase lysosomal trafficking [207]. In mice expressing L444P-GCase, oral administration of isofagomine increased the GCase levels in the brain and other tissues by two- to five-fold [207]. Of note, isofagomine improved motor and non-motor outcomes, reduced α-Syn aggregates, and reduced neuroinflammation in α-Syn-overexpressing mouse model wild-type for GBA mutations [208]. This latter small molecule chaperone showed increased GCase activity in peripheral blood cells and was extensively tested in healthy individuals and GD1 patients (ClinicalTrials.gov IDs: NCT00875160; NCT00813865; NCT00446550; NCT00433147; NCT00465062) [209]. However, further trials were stopped due to poor clinical efficacy. 

Recently, another compound has gained significant therapeutic attention: Ambroxol. This is commonly used as an oral mucolytic agent and to treat hyaline membrane disease in newborns, which has been recently demonstrated to be effective as GCase chaperone [177]. Preclinical studies showed that Ambroxol acts as a Ph-dependent, mixed-type chaperon of GCase, whose inhibitory activity is maximal in neutral Ph in the cytoplasm, intermediate inside the endoplasmic reticulum, and undetectable in the acidic Ph of lysosomes [210]. In normal and GBA-mutated fibroblasts and N370S/N370S lymphoblasts, Ambroxol increased lysosomal GCase activity and protein levels [210,211,212,213], favoring proper GCase folding within the endoplasmic reticulum and its shuffling into the lysosome [211], and reduced oxidative stress [213] with low cytotoxicity [212]. GCase activity restoration was obtained through an increase in Saposin-C and LIMP2 protein levels, thus confirming that Ambroxol acts by modulating lysosomal function and trafficking [214]. 

In mice models receiving progressively increasing doses of Ambroxol, significantly higher GCase concentrations were found in the spleen, heart, and cerebellum without significant adverse events [212]. Similarly, Ambroxol increased brain GCase levels and decreased total and phosphorylated α-Syn amounts in wild-type mice, L444P-mutated mice, and mice overexpressing human α-Syn, without toxicity [215]. The same results were observed in non-human primates [216]. In humans, the first studies to assess the safety and efficacy of Ambroxol were performed on patients with GD [216]. In this population, high-dose Ambroxol showed good safety and tolerability as well as a significant increase in lymphocyte GCase activity, reduction in CSF glucosylsphingosine levels, and clinical outcomes’ improvement [217,218,219]. 

In PD patients, high-dose Ambroxol was investigated in a phase 2a prospective, single-center, open-label non-controlled clinical trial (ClinicalTrials.gov ID NCT02941822, Aim-PD trial) [220]. Seventeen out of twenty-three moderate PD patients (eight patients with GBA mutations and nine without GBA mutations) received an escalating dose of oral ABX up to 1.26 g per day over 6 months, without serious adverse events. In all patients, CSF level of Ambroxol significantly increased during the treatment period and was significantly associated with a decrease in CSF GCase activity. Moreover, Ambroxol was associated with a significant increase in CSF α-Syn concentration and, clinically, a reduction in mean MDS-UPDRS part III score [220]. These results are very promising and paved the way for double-blind, placebo-controlled trials in larger populations of PD carriers of GBA mutations. 

Ambroxol is under evaluation in a phase 2, single-center, double-blind, randomized placebo-controlled trial in patients with mild to moderate PDD (ClinicalTrials.gov ID: NCT02914366) [221]. In the announced study protocol, 75 patients were randomized to receive Ambroxol high dose (1050 mg/day), or low dose (525 mg/day), or placebo. Clinical, biomarker (imaging and CSF measures), pharmacokinetic, and pharmacodynamic (Ambroxol plasma levels and GCase activity in lymphocytes) assessments have been performed at baseline, at 6 months, and at 12 months from recruitment [221]. This trial has been concluded and its results are yet to be published.

More recently, a phase 2, multicenter, double-blind, randomized, placebo-controlled study started recruiting a minimum of sixty patients diagnosed with PD and carriers of GBA mutations, who will be recruited and randomly allocated to either oral Ambroxol 1.2 g/day or placebo (ClinicalTrials.gov ID: NCT05287503). Given the prominent impact of GBA mutation on the risk of incident dementia, the primary objective is demonstrating a reduced progression of cognitive dysfunction over the 12-month period. Safety, tolerability, pharmacokinetic, and pharmacodynamic measures as well as motor and nonmotor variables will be assessed.

Finally, two trials of Ambroxol are in the recent pipeline for patients with DLB. A phase 1–2 randomized, placebo-controlled, double-blind study investigating its safety, tolerability, and efficacy was started in 2020 and it is expected to conclude in 2023–2024 (ClinicalTrials.gov ID: NCT04405596). A phase 2a multicenter, randomized, controlled, double-blind trial is currently recruiting new and early patients with prodromal and mild DLB, to assess Ambroxol clinical efficacy on cognitive, neuropsychiatric, and functional outcomes (ClinicalTrials.gov ID: NCT04588285, ANeED Study). An estimated study population of 172 participants will be randomized to receive five escalation doses of Ambroxol up to 1260 mg/day or placebo for 18 months. 

Regarding non-inhibitory chaperones, the NCGC607 compound was able to chaperone mutant GCase into lysosomes, restore GCase enzymatic activity and protein levels, and reduce abnormal substrate accumulation in both iPSC-derived macrophages and dopaminergic neurons from carriers of GBA mutations with or without parkinsonism [222]. In addition, NCGC607 reduced α-Syn levels in dopaminergic neurons derived from patients with PD [222]. Similarly, another non-inhibitory small molecule chaperone (named NCGC00188758) promoted GCase activity in lysosomes, reduced glucosylceramide levels, and increased the clearance of pathological α-Syn in cell lines derived from PD noncarriers and PD patients carrying mutations in SNCA (triplication or A53T), GBA, or PARK9 genes [223]. Interestingly, this compound was able to reverse cellular pathology induced by abnormal α-Syn accumulation [223]. In humans, another non-inhibitory small molecule compound (LTI-291) was tested in PD carriers of GBA mutations, showing positive safety, pharmacokinetic, and pharmacodynamic data but failing to demonstrate any benefit on neurocognitive outcomes [224]. Further studies are needed to clarify the therapeutic potentials of small molecule chaperons in PD carriers of GBA mutations.

#### 4.2.3. Gene Therapy

A second strategy that can be used to modulate GCase enzymatic activity is gene therapy. This is an exciting therapeutic strategy based on the use of viral vectors to insert wild-type alleles into the genome of mutated individuals, to correct gene abnormalities and restore normal protein function and levels. As concerns GBA, some promising results in gene therapy have been achieved in preclinical studies. In rodent models of PD overexpressing wild-type α-Syn, intra-cerebral multi-sites injections of AAV-GBA increased GCase activity and reduced α-Syn levels in the SN and striatum [225], whereas in models induced by AAV-A53T mutant α-Syn, co-injection into the SN of AAV-GBA with AAV-A53T α-Syn protected dopaminergic neurons against neurodegeneration [225]. Similarly, in A53T SCNA transgenic mice, AAV-PHP.B-GBA1 administration restored normal enzymatic levels, reduced α-Syn pathology, and improved behavioral outcomes [226]. Benefits from gene therapy were also observed in animal models of GD, as lentiviral vector-mediated delivery of normal GBA alleles corrected GD phenotype [227,228]. 

More recently, two trials with GBA1 gene therapy (PR001) in GD2 and in GBA-associated PD have started. The former PRV-GD2-101 (ClinicalTrials.gov ID: NCT04411654, PROVIDE Trial) is an open-label, Phase 1/2, multicenter study to evaluate PR001 intracisternal administered single-dose safety, efficacy, immunogenicity, and biomarkers in infants with GD2. The latter PRV-PD101 (ClinicalTrials.gov ID: NCT04127578, PROPEL Study) is a phase 1/2a multicenter, opening-label, ascending dose, first-in-human study to evaluate the safety, tolerability, immunogenicity, and clinical effects of intracisternal high-dose and low-dose PR001 administration in patients with moderate-severe PD carrying a GBA mutation. These two studies are expected to be completed in 2028 and 2027, respectively.

### 4.3. VPS35 (Vacuolar Protein Sorting 35, OMIM*601501)

A missense mutation of the Vacuolar Protein Sorting 35 (VPS35) gene was identified as a possible cause of autosomal dominant PD in 2011 in PD pedigrees of Swiss and Austrian origin [229,230]. Mutations of VPS35 are very rare, being recognized in almost 0.2% of autosomal dominant PD cases in Europe [21]. The most common mutation is D620N (OMIM*601501.0001), clinically manifesting in the fifth decade, with a tremor-predominant phenotype, and levodopa-responsive parkinsonism [21,230]. VPS35 gene encodes for a component of the multimeric retromer complex, which is primarily involved in the modulation of endosomes trafficking [21]. Several studies linked VPS35 to α-Syn metabolism, dopamine neuron functions, and survival. Indeed, VPS35 acts in macroautophagy, chaperone-mediated autophagy, lysosomal pathway promoting α-Syn clearance, in transport and localization controls of the protein involved in α-Syn degradation [231]. In addition, as a component of a multimeric retromer complex, VPS35 contributes to proper synaptic function through regulation of synaptic plasticity, protein, and vesicular trafficking, and dopamine transporter recycling in dopaminergic neurons [231].

Considering that VPS35 mutations lead to abnormal retromer complex function [230], therapeutic strategies are based on the stabilization of the complex. In cellular models, chemical chaperones improving retromer complex stability increased the levels of retromer proteins, including VPS35, promoted the shift of amyloid precursor protein (APP) from the endosome, and decreased APP-induced pathological changes [232]. On the other hand, restoring the normal VPS35 gene expression through AAV-mediated gene therapy—as currently explored for other PD-related genes—might be useful in the early PD stage to prevent dopaminergic neurons loss [233]. Finally, it cannot be excluded that therapeutic approaches tested for other genetic forms of PD could be either beneficial in VPS35-related forms, due to overlapping molecular mechanisms [231]. 

## 5. Autosomal Recessive Genes 

### 5.1. Parkin (PRKN or PARK2, OMIM*602544)

Mutations in the gene encoding for Parkin (PRKN) are the most frequent cause of autosomal recessive PD, with a prevalence of 10–20% in early onset PD (age at onset < 40–50 years) [234]. Several mutations in the PRKN gene have been reported, from missense mutations to exon rearrangements, which result in different biochemical consequences on protein function [21,234]. 

Clinically, homozygous or compound heterozygous mutations in the PRKN gene cause a predominantly early-onset slowly progressive parkinsonism, with diurnal fluctuation, dystonic features mainly involving the lower limbs, and good levodopa response, which is frequently complicated by Levodopa-induced motor fluctuations and dyskinesias [47]. Overall, non-motor symptoms are less severe in PARK2-associated PD, but neuropsychiatric disturbances could be prominent [47]. 

PRKN protein is an E3-ubiquitin ligase involved in mitochondrial biogenesis, fusion/fission, and transport, and regulates mitophagy and endoplasmic reticulum–mitochondrial interactions [234]. In addition, PRKN contributes to the regulation of cellular proteostasis through the UPS [234], pro-survival NF-kB pathway, cell cycle, synaptic function, and vesicular trafficking [235]. 

Due to its central role in mitochondrial homeostasis, preclinical studies have mainly focused on discovering and testing compounds able to rescue normal mitochondrial function in PARK2-mutated cells. In a large screening of more than 2000 compounds in PARK2-mutated fibroblasts, almost fifteen exhibited potential mitochondrial rescue abilities, of which two compounds (ursocholanic acid and ursodeoxycholic acid) were selected for further examination [236]. In PD animal models, ursodeoxycholic acid (UDCA) reduced rotenone-induced apoptosis, halted striatal dopaminergic cell death, and improved motor performances [237]. From bench to bedside, several clinical trials have been promoted to explore UDCA effects to modify disease progression. In a small open-label trial on five PD patients, UDCA showed safety, tolerability, and variable non-linear pharmacokinetics. Effect on brain metabolism was evaluated only in three out of five patients, showing a modest increase in ATP and decrease in ATPase activity, but the small sample size provides unsecure results [238]. 

A subsequent phase 2 randomized, placebo-controlled, double-blind trial (“UP Study”; ClinicalTrials.gov ID: NCT03840005;225)] enrolled thirty PD patients not carrying PRKN mutations. The patients were randomized 2:1 to receive UDCA 30 mg/kg or placebo for 48 weeks, followed by an 8-week washout phase. The study was concluded in May 2021 and results are expected [239]. Another ongoing trial investigating the effect of UDCA on brain bioenergetics using 7 Tesla-Magnetic Resonance Spectroscopy (MRS) began in 2022, and aims to correlate UDCA pharmacokinetics measurements to cortical bioenergetics’ profile measure and ATPase levels (ClinicalTrials.gov ID: NCT02967250). 

To our knowledge, no trials on UCDA are currently ongoing specifically in PARK2-related PD. 

Another potential therapeutic strategy in PRKN mutation PD patients might be Rapamycin. Indeed, Parkin can directly interact with mTOR and this interaction promotes proper mTOR activity under oxidative stress [240]. In preclinical studies, rapamycin-mediated activation of the translation inhibitor 4E-BP abolished degeneration of dopaminergic neurons and ameliorated mitochondrial defects in cells derived from PRKN-mutant PD patients [241]. 

### 5.2. PINK1 (PTEN-Induced Kinase 1, OMIM*608309)

PINK1 gene mutations are the second most common cause of autosomal recessive PD after PRKN. The gene encodes for the mitochondrial kinase PTEN-induced kinase 1 [242]. PINK1 mutations have been confirmed in multiple ancestries, and a 4–7% mutation frequency is currently reported among sporadic early-onset PD [47]. The clinical phenotype resembles PRKN-related PD with typical slow-progressive parkinsonism with good and persistent response to Levodopa and low prevalence of non-motor symptoms [47]. The atypical features are dystonia, sleep benefit, and hyperreflexia [47]. Similar to PRKN, PINK1 contributes to the maintenance of mitochondrial homeostasis and regulates UPS function [21,243]. PINK1 mutations induce a complete loss of kinase activity, mitochondrial dysfunction, oxidative stress, impaired cellular metabolism, abnormal mitophagy, and altered proteostasis [21,47]. In this scenario, Parkin and PINK1 could share similar therapeutic strategies and further research is warranted. 

### 5.3. DJ-1 (Parkinsonism-Associated Deglycase, OMIM*602533)

The DJ-1 gene is identified as responsible for a rarer autosomal recessive form of PD [244]. DJ-1 mutations are responsible for <1% of early-onset PD, clinically manifesting as Levodopa-responsive parkinsonism, associated with dystonic and pyramidal features [46]. The exact protein function remains unknown, but shreds of evidence suggest its role in mitochondrial function, cell oxidative-stress response, and transcriptional regulation [21,244]. Moreover, DJ-1 protein interacts with PRKN and PINK1 to form a ubiquitin ligase complex, whose function is compromised by pathogenic mutations [21]. Several compounds have been evaluated in preclinical models as potential therapeutic targets in PD, including recombinant DJ-1 protein [245], AAV-DJ-1 [245], and the short DJ-1-based peptide ND-13 [246], and DJ-1-binding compounds UCP0054277 andUCP0054278 [247,248,249,250,251]. These compounds prevented TH-positive neurons’ cell death and suppressed motor dysfunction (for more details see [244]). In in vitro and in vivo models of PD, 11-Dehydrosinulariolide increased cytosolic and mitochondrial DJ-1 expression, reversed TH-positive neurons damage, and ameliorated motor functions [252]. Similarly, the safflower flavonoid extract restored the expression of TH, dopamine transporter, and DJ-1 protein, increased dopamine levels, and improved behavioral and motor parameters in rotenone-induced rat models [253]. 

As for previously mentioned autosomal recessive genes, no clinical trials are currently ongoing targeting the DJ-1 pathway. 

## 6. Challenges of Genetic-Driven Disease-Modifying Therapies

Disease-modifying strategies on PD stratified according to the molecular abnormalities caused by mutations in PD-causing genes are unquestionably an exciting strategy to promote a model of precision medicine. Genetic screening could provide a ‘pathophysiological fingerprint’ driving targeted therapeutic approaches. It is worth emphasizing that genetic characterization provides insights into molecular mechanisms that could be therapeutically targeted even in sporadic PD and hypothetically enables therapeutic interventions in prodromal or asymptomatic carriers of genetic mutations. In this perspective, the genetic characterization of PD could help encompass an important shortcoming of disease-modifying strategies, namely the assumption that all patients with clinical diagnosis of PD present the same disease. Nonetheless, researchers need to deal with several challenges to grab this opportunity. Currently, no animal models properly resembles the α-Syn pathology that occur in PD [254]. This severely limits the reliability of preclinical trials for known compounds and the identification of potential future therapeutic candidates. Although preclinical models (transgenic mouse models; viral-vector-based animal models; ‘prion-like’ models; chemically induced damage to dopaminergic neurons by 1-methyl 4-phenyl-1,2,3,6 tetrahydropyridine, 6-hydroxydopamine, or rotenone) have proven useful in testing symptomatic therapeutic strategies, they have a much more limited assessment of disease-modifying strategies due to the different pathologies that the models show compared to PD. Indeed, the acute degeneration of dopaminergic neurons in the preclinical model does not reflect α-Syn pathology and neurodegeneration [254]. In addition, transgenic models rarely express α-Syn containing post-translational modifications found in PD (e.g., phosphorylation or C-truncation) that can contribute to α-Syn toxicity and, on the other hand, the overexpression of α-Syn shows poorly predictable SN neurodegeneration and high motor phenotypes’ variability [254]. Some advantages to mimic neuropathology progression in PD are shown in prion-like models, derived from the injection of α-Syn fibrils into the brain [254]. Despite these recent advances, an intrinsic limitation of all animal models remains their short lifespan when compared to the long disease duration of human PD. 

A second challenge regards the absence of validated biofluid-based, α-Syn neuroimaging or clinical biomarkers to assess prodromal stages, disease severity, progression, target engagement, and treatment responses. Validated biomarkers are fundamental in all steps of therapeutic trials: correct diagnosis; patient selection, subtyping, and stratification; target engagement; recognition and measure of disease-modifying effects; quantification of drug impact on disease progression, motor/non-motor outcomes, quality of life, and survival. Lack of biomarkers or a wrong biomarkers’ selection may lead to considering a compound non-effective, only because its effects are not recognized or because it is tested on the wrong patients and/or at the wrong time. Moreover, biomarkers are needed to clarify if a measured drug-induced change really exerts a disease-modifying role. Consequently, a further intrinsic challenge is to define which can be considered a ‘correct’ biomarker, requiring a wide knowledge of disease pathogenesis, biochemical mechanisms, and neuropathology. 

To date, although enormous advances in understanding PD pathology have been achieved, some of the aspects still remain equivocal. For instance, the role of α-Syn in triggering PD pathology is quite clear in specific genetic forms of PD resulting from mutations in the SNCA gene or α-Syn overexpression (misfolding proteins aggregate and propagate through a prion-like mechanism) [25,26,27,28,29,30]; this is not the same in sporadic PD, where α-Syn pathology could be considered a compensatory, or even a protective, effect [6]. However, it could not be excluded that SNCA gene mutation might reduce or annihilate a protective effect of α-Syn [6]. These statements might be extended to all the other pathogenic mechanisms implicated in PD (e.g., lysosomal and autophagic dysfunction, mitochondrial damage, synaptic dysfunction, neuroinflammation, etc.). Are these mechanisms secondary to α-Syn pathology or do they represent the *‘primum movens’* of PD pathogenesis? α-Syn species show a different pathogenic role, and it is unclear if α-Syn post-translational modifications can affect pathogenicity or antibodies’ affinity. Additional open questions are also concerned with LBs’ pathology: nigrostriatal cell death occurs very early when LBs are not detectable [6]; LBs do not correlate with symptom severity, and LBs can be incidentally found in the brains of healthy elderly individuals [255,256] or be even absent in some genetic forms of PD [257,258]. Moreover, recent evidence suggests that α-Syn oligomeric species are more toxic than its more aggregated forms [259,260,261]. In accordance, α-Syn oligomers differ from LBs in their topographical distribution—the former being more diffuse in the neocortex and the latter more abundant in the brainstem—they are not found in control subjects and correlate with severity of cognitive impairment [262]. 

All of these considerations not only question the classic pathogenetic model of a toxic gain-of-function synucleinopathy, which assumes that toxic α-Syn aggregates are per se triggers and promoters of disease progression, but also suggest the alternative fascinating hypothesis that the loss of physiological endogenous α-Syn, due to its recruitment first into oligomers and then into aggregates, could actually lead to toxicity via a loss-of-function mechanism. Compelling evidence shows that α-Syn contributes to dopamine neurons viability [263], regulates synaptic function homeostasis [23], and probably intervenes in immune-mediated responses [24], via interaction with a variety of cellular proteins [122]. Of note, an accurate model of nigrostriatal degeneration could be reproduced knocking down normal α -syn in preclinical models [89,263]. In these models, the neurotoxicity correlates with the degree of α-Syn downregulation and consequently with the level of residual normal α-Syn, and could be rescued by overexpressing the normal protein [89,263]. As has been recently suggested, these concepts “crack” the traditional disease paradigm of PD [255,264] and raise some concerns about the possibility that current α-Syn elimination therapies could even be deleterious due to a further reduction in normal α-Syn. In this perspective, enhancing strategies aimed at maintaining or augmenting levels of soluble normal α-Syn should be advocated instead [255,264].

Another challenge comes from the usually brief duration of clinical trials. Due to slow disease progression, particularly in some genetic forms, long follow-ups could be necessary to recognize any effect on disease progression and biomarkers. 

Finally, it is worth noting that genetic-driven clinical trials impose even more precise criteria in patient selection, as some biochemical mechanisms could be strictly mutation-specific. 

## 7. Role of Genetics in Decision-Making Process on Symptomatic Therapies 

A clear and dramatic beneficial response to dopaminergic therapy is considered a supportive diagnostic criterion in the current MDS Diagnostic criteria for PD [265]. However, clinical experience shows high interindividual variability in drug response, clinical benefits, development of motor complications, and drugs’ side effects. This variability is probably multifactorial, depending on complex interactions between environmental, epigenetic, and genetic factors [266,267]. Specifically, polymorphisms in genes involved in dopamine synthesis and metabolisms, such as catechol-O-methyltransferase (COMT), monoamine oxidase-B (MAO-B), dopa decarboxylase, Dopamine receptors, and Dopamine transporter genes, influence the amount of Levodopa intake, clinical response to the drug, risk of developing motor fluctuations, dyskinesias, and drug-induced adverse reactions (hallucinations, psychotic episodes, impulse control disorders, gastrointestinal adverse effects, sleep attacks) [266,267]. Similarly, several variants in the LRRK2 gene have been associated with the risk of motor fluctuations and levodopa-induced dyskinesias [267,268]. 

Despite the characterization of polymorphisms remaining quite limited in research settings, genetic phenotyping may provide useful clues not only for access to disease-modifying strategies but also for potentially predicting responses to dopaminergic drugs and guiding the most suitable therapeutic choice, based on the estimated rate of response and the expected motor and non-motor outcomes. Nonetheless, more systematic study protocols on wider populations are required to reach conclusive results, as the majority of available data, mainly coming from systematic reviews and meta-analysis of single case reports or case series, are limited by small sample sizes, incomplete data, and heterogeneous methodology. For instance, merely qualitative criteria and high variable cut-offs are often used to define the response to pharmacological and device-aided therapies [269,270], making the interpretation of data complex and arbitrary.

## 8. How Genetic Status Can Help in the Current Clinical Management of PD 

### 8.1. SNCA

As previously mentioned, SCNA mutation carriers display an earlier age at onset, and more rapid motor and non-motor progression [47,48]. Nevertheless, different SCNA mutations manifest variability in clinical phenotype. SCNA p.A53T and p.E46K missense mutations induce early-onset parkinsonism associated with cognitive impairment. Similarly, carriers of SNCA duplications display early parkinsonism with more rapid cognitive deterioration, dominated by initial attentive/executive dysfunction and subsequent decline in visuospatial abilities, resembling DLB [271]. SCNA gene triplications cause an even more severe non-motor phenotype not only in the cognitive domain but also in psychiatric (depression, psychosis) and autonomic (gastrointestinal-urinary disorders and cardiovascular autonomic failure) symptoms [46,49]. Conversely, other non-motor symptoms, such as anxiety and sleep disturbances, are similarly represented in all patients, with no clear differences according to the mutation type [46]. A genotype-phenotype systematic review of 146 patients carrying SCNA mutations collected in MDSGene database reported depression in most cases of SNCA triplication and in nearly 60% of patients with gene duplication; psychotic symptoms and hallucinations in 96% of duplication carriers, 83% of triplication carriers and 20% of the missense mutations carriers; dementia in 67% of missense mutation carriers, 88% of SNCA triplication carriers; autonomic failure in 60% of triplication carriers, 41% of duplications carriers, and 48% of patients with missense mutations [48]. In the MDSGene database, a therapeutic response to dopaminergic drugs was reported among all patients carrying SNCA, with “good” and “sustained” benefit in the majority (78%), in accordance with high prevalence of dopamine-responsive symptoms within the cohort (99% of patients with bradykinesia, 100% with rigidity, 88% with tremor) [48]. Nevertheless, data were insufficient to precisely quantify the rate of response and nearly half (44%) of the patients with good Levodopa responsiveness developed motor fluctuations and dyskinesias [47]. In addition, the coexistence of atypical non-dopaminergic signs (89% with postural instability and 83% with dystonic/pyramidal features), precluded a complete motor response to dopaminergic therapy [48]. Over et al. reported similar results in a more recent study [272]. Among 82 out of 146 SNCA carriers for whom treatment information was available, 79.3% of patients displayed a good response on relative low mean doses of Levodopa, nearly 10% showed a moderate response to higher doses, and 11% had a minimal response [272]. In both studies, no obvious differences emerged according to mutation type, but good levodopa responsiveness was more frequently observed in SNCA duplication and triplication carriers compared to p.G51D mutation carriers, despite a higher rate of motor and non-motor adverse events (dystonia, dyskinesias, hallucinations, psychosis) [48,272].

As regards non-levodopa medications, dopamine agonists, COMT inhibitors, MAO-B inhibitors, anticholinergic medications, or NMDA antagonists evidenced benefit in most patients [272]. Both dopaminergic and non-dopaminergic drugs (e.g., amantadine, anticholinergics, tricyclic antidepressants) contribute to non-motor symptoms worsening, due to potential exacerbation of psychiatric disorders, impulse control disorders, autonomic dysfunction, and cognitive decline, commonly in SNCA carriers. 

Finally, the good response to levodopa, the tendency to develop dystonia, dyskinesia, and motor fluctuations, make PD carriers of SNCA mutations good candidates for device-aided therapy. However, other common symptoms such as postural instability, cervical dystonia, pyramidal signs, and alien limb phenomenon, do not improve by device-aided therapy, and cognitive decline and psychosis are exclusion criteria for some strategies such as DBS and Continuous Apomorphine Subcutaneous Infusion (CASI) [49]. 

*Device*-*aided therapies*. Limited data are available on the relationship between SNCA mutation and response to device-aided therapies. To date, literature reported eight patients carrying SNCA mutations received either bilateral subthalamic nucleus deep brain stimulation (STN-DBS) (*n* = 5), bilateral Globus Pallidus internus (GPi) DBS (GPi-DBS) (*n* = 1), or thalamotomy/pallidotomy brain surgery (*n* = 2) [270,272,273,274,275,276,277,278,279,280]. All patients displayed a beneficial motor response to DBS, sustained over a follow-up period ranging from 1 months to 4 years [270,272,273,274,275,276,277,278,279,280]. Among cases of SNCA duplication, Antonini et al. reported a 43% improvement in UPDRS III, 87.5% reduction in motor complications, and almost 50% reduction in LED at one year after surgery [281]. Similarly, Shimo et al., and Elia et al. reported reduction of UPDRS III ranging from 42% to 52% and a LED reduction from 29.7% to 58% four and three years after surgery, respectively [280,282]. Ahn et al. reported an “excellent” motor response to bilateral STN-DBS but the amount of the response was not quantified [278]. Only one case carrying p.A53T SNCA mutation has been described to date with a 1-year post-surgery follow-up, showing a 43% motor improvement, along with a 63% reduction of LED [276]. A recent multicenter retrospective study collected data on previously reported patients, with a longer follow-up (up to 10 years) [273]. Three patients carried SNCA duplication and one missense p.A53E mutation. All patients with SNCA duplication showed a favorable long-term motor response to DBS [273,281,282], with improvement in motor complications up to 87% [273,281]. In particular, all three patients carrying SNCA duplication displayed substantial stability or only minor increase in the UPDRS III score at last follow-up up to 10 yrs from DBS compared to baseline, maintained a stable reduction in UPDRS items for motor complications, total daily LED, and number of Levodopa doses per day [273]. Conversely, despite initial good response, the patient carrying the missense mutation became wheelchair-bound due to motor axial progression (UPDRS III score at baseline: 10; at follow-up: 56; H&Y stage at follow-up: 5) and showed poor cognitive and psychiatric outcomes 3.5 years after surgery [273]. Similarly, despite satisfactory motor outcomes, the majority of patients with SNCA duplication receiving STN-DBS displayed substantial cognitive and psychiatric deterioration [269,276,279,282].

Data on outcomes after GPi-DBS are limited. One patient carrying mosaicism of SNCA duplication underwent bilateral pallidal stimulation for a dyskinetic-dystonic predominant phenotype, exhibiting clinically relevant improvement and complete abolition of peak-dose dyskinesias one month after surgery [277]. 

Current evidence is even more limited for other device-aided therapies (CASI and Levodopa-Carbidopa Intestinal Gel (LCIG)) in SCNA-related PD [283], but higher frequency and earlier development of dementia, hallucinations, psychosis, and autonomic dysfunction could make these approaches unsuitable as well. 

Based on these findings, STN-DBS could be beneficial for motor fluctuations in SCNA-PD, despite a higher long-term risk of axial motor decline and non-motor (cognitive and psychiatric) progression [273,283]. Similarly, other non-surgical device-aided therapies, such as CASI, could exacerbate non-motor complications of dopaminergic stimulation. In these patients, LCIG seems a potentially more suitable and safe therapeutic option [283]. 

### 8.2. LRRK2

LRRK2-related PD strongly resembles idiopathic PD [49,143,144,150]. Overall, the majority of LRRK2-PD patients display a good response to Levodopa [48,49,272], with a rate of motor complications similar to that of PD noncarriers [151]. In a recent systematic literature review assessing 820 LRRK2 mutation carriers, among which 545 were receiving Levodopa therapy with a mean treatment duration of 17 years, almost 95% of patients displayed a good motor response, albeit with a highly variable LED, 5 patients had a moderate response (mean LED 600 mg/day), and 15 patients displayed only minimal clinical benefit [272]. Treatment with dopamine agonists and other dopaminergic medications is generally beneficial, without any significant adverse event [272]. Among responsive patients, almost 35% of LRRK2 mutation carriers developed dyskinesias and motor fluctuations, more severe and frequent in p.R1441C/G/H/S mutation [48,49].

*Non*-*motor symptoms*. Most studies suggest that LRRK2-related PD displays less non-motor symptoms compared to PD noncarriers: 23% and 32% presented cognitive decline and psychiatric disturbances, respectively [49,144,151]. LRRK2 p.G2019S mutation carriers can manifest higher propensity to daytime sleepiness and sleep attacks [274], which should be taken into account when considering dopamine-agonists’ administration.

*Device aided*-*therapies*. To date, a larger number of LRRK2 patients have received brain surgery, DBS, pallidotomy, or thalamotomy, compared to other monogenic PD forms [269,270,272,275,279]. Good motor response and improved motor complications were reported for the majority of the patients, despite variable follow-up duration ranging from 3 months to 7 years and some genetic variability [269,270,275,279]. 

In particular, de Oliveira et al. reported outcomes of 50 LRRK2 carriers (44/50 carrying p.G2019S mutation) treated with bilateral STN-DBS [269]. Among the patients with a shorter follow-up, improvement was qualified as “marked” or “satisfactory” in nearly 46% of patients, while “unsatisfactory” outcomes were reported for less than 9% of patients [269]. Among patients with intermediate follow-up (58%, 2–6 years), the totality of patients displayed either “marked” or “satisfactory” motor response, while all the patients with long-term follow-up maintained a “sustained” improvement [269]. Globally, motor complications’ improvement ranged from 33.3% to 75%, and LEDD was reduced from 17.5% to 75%. Interestingly, the best outcome was achieved by patients carrying either p.G2019S, p.T2031S, or p.Y1699, whereas p.R144G mutation carriers displayed a less satisfactory response [269,284]. Similarly, among 79 LRRK2 carriers from seventeen studies undergoing STN-DBS and followed up to 7 years, motor outcomes were favorable in the majority of patients, while p.R144G mutation carriers had poor outcomes [270]. Nonetheless, quite different cut-offs to quantify motor response to DBS were used in the two above-mentioned systematic reviews [269,270]. More recently, a surgical cohort of 46 LRRK2 patients has been described in a systematic review and meta-analysis [275]. UPDRS III scores improved by 46% in LRRK2 carriers compared to 53% in idiopathic PD and UPDRS IV improved by 50% to 75% [275]. Mean LEDD was reduced by 61% compared to 55% in non-carriers [275].

In longer follow-ups, p.G2019S carriers have been reported to present even better DBS motor responses compared to PD non-carriers, maintained up to 10 years, despite some progression in axial signs [285,286]. Regarding non-motor symptoms, cognitive performance remained stable up to almost 6 years after surgery, without cognitive decline reported, and psychiatric complications (behavioral disorder and hallucinations) occurred only in p.T2031S variant carriers. Globally, disability and quality of life showed good or minor improvement [269,284,286]. These data suggest that LRRK2 mutation carriers are good candidates for STN-DBS when commonly used inclusion criteria are met. 

For other device-aided therapies, experience is more limited. Within a cohort of 12 PD patients on LCIG, Foltynie et al. described one LRRK2 carrier treated with LCIG after a 19-year history of PD and a previous unsuccessful attempt with DBS. The patient did not display significant improvement in daily motor diary and quality of life and died after 24 months from colon cancer [287]. More recently, a study compared motor responses to LCIG between 5 LRRK2 carriers and 17 non carriers within a cohort of 44 PD patients [288]. No significant differences were found among the groups [288]. Similarly, an abstract published from the same study group reported no differences in motor response to LCIG between 16 LRRK2 carriers, 11 GBA-PD, and 42 idiopathic PD, even though motor UPDRS scores were significantly higher in GBA-PD group [289]. Dyskinesias were reported in 93% of LRRK2 carriers (compared to 90.6% of noncarriers) and nearly 43% displayed hallucinations compared to 63.6% of non-carriers [289]. Only one LRRK2 carrier has been treated with CASI to date, but the motor and non-motor outcomes were not specified [290]. 

### 8.3. GBA

Heterozygous GBA mutations are responsible for a more aggressive clinical phenotype, characterized by an early age at onset, rapid motor and non-motor progression, and reduced survival, especially in carriers of severe GBA mutations [177,181,190]. 

*Motor symptoms*. Dopaminergic motor symptoms prevalently present asymmetric akinetic-rigid phenotype onset [181,182,188,189,190,291,292], excellent motor response to Levodopa associated with greater risk and earlier appearance of motor fluctuations and dyskinesia [170,173,174,175,176,277,278,279]. 

Nonetheless, despite similar UPDRS III scores among GBA carriers and non-carriers [189,191,292,293], the H&Y stage and Levodopa-non responsive symptoms are significantly higher in GBA carriers compared to PD noncarriers [189,191], postural instability/gait phenotype (PIGD) is more frequent in GBA-PD [189,191], and the carriers of severe GBA mutations display more severe motor symptoms and H&Y stage OFF-medication [191]. 

A quantitative motor response to conventional dopaminergic treatment is not provided in the current available literature [181,182,188,189,190,191,291,292], except for a recent retrospective study comparing baseline UPDRS III score, response to Levodopa Challenge Test (LDC), and best ON-medication UPDRS III score among 13 early GBA-PD compared with 48 sporadic PD [294]. The study shows a similar motor response to LDC and standard oral Levodopa therapy among groups [294]. 

Compared to PD noncarriers, GBA carriers display a faster motor decline subsequent to a steep deterioration of non-dopaminergic axial features, which substantially contributes to a poor quality of life and increases the risk of institutionalization and falls [190,295,296,297]. 

*Non*-*motor symptoms*. Besides the motor phenotype, non-motor symptoms are prominent in GBA carriers as well, both in symptomatic and prodromal phases [298,299,300]. GBA carriers suffer from severe autonomic dysfunction (orthostatic hypotension; urinary, sexual, and bowel dysfunction, and sweating), sleep disorders, neuropsychiatric disturbances (anxiety, depression, apathy, visual hallucinations, and other psychotic symptoms), and cognitive decline [190,294,295,296,301,302,303,304,305,306]. It is noteworthy that GBA carriers have an increased mortality risk compared to PD non-carriers partially independent of age and dementia [190], suggesting that other factors (including orthostatic hypotension and/or other non-motor symptoms), could negatively affect survival in these patients. Overall, GBA-related PD has a complex clinical spectrum that should be carefully considered to target therapeutic choices. Caution should be applied with the use of dopamine-agonists or anticholinergic drugs for their potential risk to worsen autonomic and/or cognitive functions as well as psychiatric disturbances. 

*Device*-*aided therapies*. Recent studies explored the outcome of DBS in GBA-PD by investigating the prevalence of GBA mutations in surgical PD cohorts and by exploring motor and non-motor outcomes in GBA carriers treated with DBS. Due to the high risk of motor fluctuations and dyskinesias, GBA-PD could be reasonably overrepresented in DBS cohorts (mostly STN-DBS). A variable prevalence ranging from 3% to 17% of PD patients who underwent DBS resulted in being carriers of a GBA mutation [307,308,309]. Globally, the available literature outcomes were obtained from 74 GBA carriers (2 patients also carried LRRK2 mutation [310,311] and 1 patient was heterozygous GBA plus PRKN mutation [311]), among which 58 patients were treated with STN-DBS, 4 with DBS of the GPi, 1 patient with DBS of the Vim, and 11 patients with not-specified targets [307,308,309,310,311,312]. Good motor response to STN-DBS has been shown in the short term, with a reduction of 40% in UPDRS III score Stim-On/Med-Off vs. presurgical Med-Off [309], confirmed in a p.N370S mutation carrier (almost 33.3% from Med-Off/Stim-Of to Med-Off/Stim-On condition) and in two p.L444P carriers (varied from 20% to 89%) [306]. 

However, this initial benefit declined at longer follow-up, with axial deterioration occurring [310,311] and increasing rigidity sub-scores [311]. The latter results were not reported by Weiss and colleagues who reported a reduction in UPDRS III score from Med-Off/Stim-Off to Med-Off/Stim-On conditions, ranging from 36.3% to 62.5% in three GBA carriers evaluated up to 10 years after surgery [306]. 

Similarly, the benefits from STN-DBS on motor complications, assessed with UPDRS IV score, ranged from 37% [310] to 100% [307]. Less data are available on the outcome of other DBS targets in GBA-PD. Five GBA carriers undergoing DBS of the GPi (*n* = 4) and Vim (*n* = 1) have been described to date, showing an overall safety profile and variable outcome in terms of improvement of motor disability and reduction in dopaminergic medications [310,311]. In particular, only one study reported clear motor outcomes for these targets [310], showing a 22% and a 43% improvement in UPDRS III score compared to pre-surgical score in GPi-DBS and Vim-DBS patients, respectively [310]. A significant reduction of UPDRS IV score (−94%) was observed in the two patients treated with GPi-DBS [310].

Currently, the non-motor outcomes after DBS in GBA carriers raise the major concern, due to the potentially detrimental effect of STN-DBS on cognition [312,313]. Concerning cognitive performance, the majority of studies reported a faster rate of cognitive decline in GBA carriers compared to noncarriers, often associated with psychiatric complications (such as depression, anxiety, and visual hallucinations) and more severe orthostatic dysregulation [307,310,311]. In a recent multicenter study investigating clinical and genetic data of 366 subjects (58 GBA+DBS+ vs. 82 GBA+DBS− vs. 98 GBA−DBS+ vs. 128 GBA−DBS− subjects), stratified according to GBA mutation severity and longitudinally followed up to 60 months after surgery, showed a faster cognitive decline in GBA-carriers undergoing DBS compared to all other groups, suggesting a combined detrimental effect of GBA mutations and DBS on cognition [314]. Based on mutation severity, subjects carrying mild or severe mutations and undergoing DBS declined faster compared to both subjects with a non-neuronopathic GBA variant-DBS+ and GBA+DBS− patients, while there was no difference in the rate of decline between mild vs. severe mutation carriers [313]. 

As concerns non-surgical device-aided options, Thaler et al. described 11 GBA-PD patients undergoing LCIG compared to 42 PD noncarriers [289]. GBA-PD displayed higher UPDRS motor scores and required lower LCIG doses due to higher rates of hallucinations (71.4% vs. 63.6%) and cognitive disfunction compared to idiopathic PD [289]. To our knowledge, no cases treated with CASI have been reported so far. 

Taken as a whole, these data suggest the opportunity to include an assessment of GBA carrier status as part of the presurgical decision-making process and the necessity to counsel patients about the potential risks of cognitive and non-motor deterioration associated with DBS. Nevertheless, randomized, and multicenter studies on larger cohorts are advocated to clarify the effect of specific GBA mutations. 

### 8.4. PRKN 

PRKN-related PD generally display the earliest age at onset, during their 30s, with a good and sustained response to Levodopa and other dopaminergic treatments. Indeed, patients require low LEDD for an excellent control of motor signs, but despite exceedingly low doses of Levodopa, they exhibit frequent development of motor complications and dyskinesias since early disease stages of the disease [47,272,315,316]. In a large cohort of 958 PRKN patients, 18% developed dystonia, 15% motor fluctuations, and 68% Levodopa-induced dyskinesias [294]. The treatment of limb and foot dystonia may benefit from botulinum toxin and anticholinergic drugs (i.e., Trihexyphenidyl) [315,317]. These two treatment options might be useful also for axial symptoms, including camptocormia, anterocollis, scoliosis, and back pain, which are striking features of the specific parkin mutation and poor response to dopaminergic therapy. Analgesics and physical therapy are also needed to better control the axial symptoms [318]. In contrast, the only current available oral drug for the symptomatic treatment of levodopa-induced dyskinesias is Amantadine. Electrocardiography and cognition tests should be completed before the Amantadine prescription [319]. Based on the previously mentioned features of PD patients carrying the PRKN mutation, starting with a low LEDD and preferring dopaminergic treatment rather than levodopa is recommended in the early stages. However, in some studies, dopamine agonists are less effective on motor symptoms and do not truly delay the onset of motor fluctuations and dyskinesias compared to levodopa [320].

*Non*-*motor symptoms*. PRKN-related PD non-motor symptoms resemble those of PD noncarriers with less autonomic and cognitive impairment [47,49,315,321]. Orthostatic hypotension and myocardial sympathetic denervation are less pronounced in PRKN mutation carriers compared to PD noncarriers [322]. PRKN carriers display better attention [323], memory, and visuospatial abilities compared to PD [324]. Nevertheless, neuropsychiatric disturbances are generally prominent and severe in PRKN mutation carriers [325,326], mainly, impulsive-compulsive behaviors (ICB), including compulsive buying and sexual behavior, binge eating, and hobbyism/punding [326]. Psychotic disturbances are less frequent, usually occurring in dopaminergic overstimulation [325]. Risk factors for therapy-related ICB include young age, male sex, previous neuropsychiatric disturbances (anxiety, depression, apathy), and novelty-seeker impulsive personality [327]. These clinical characteristics impose careful selection of dopaminergic therapy doses, as well as frequent monitoring for impulse-control disorders and behavioral disturbances. Management of ICB is centered on reducing dopamine-agonist doses, which should be balanced with the risk of worsening motor fluctuations or causing a DA withdrawal syndrome [302]. If needed, first-line treatment options include quetiapine and clozapine. Pimavanserin has fewer side effects, but it is not approved in all countries [327,328]. 

A dominant sensory axonal polyneuropathy may occur in PRKN carriers unrelated to oral drug use or vitamin deficits. It is probably due to mitochondrial dysfunction following the Parkin mutation, and treatment with vitamin B12 supplements and COMT is not helpful in this group of patients [329].

*Device*-*aided therapies*. PRKN-related PD individuals consistently show an excellent and sustained response to DBS. Parkin PD carriers have the youngest age at DBS implant, because of the earliest age at onset and the earlier motor fluctuations’ development compared to PD noncarriers [287,316]. Forty patients, among which 22 were homozygous or compound heterozygous, have been treated with STN-DBS [330,331,332,333,334,335]. Marked motor improvement was reported by nearly 67% of patients, 17% had satisfactory responses, while the remaining showed unsatisfactory outcomes [332]. Nevertheless, most patients have short follow-up data: the ones (7 patients) with intermediate follow-up reported satisfactory responses in 57.1%, marked improvement in 29%, and poor outcomes in only 1% [268]. Unsatisfactory outcomes have been reported as higher in patients with prominent axial symptoms [269,318]. Most of the implant patients show a reduction in UPDRS-IV scores ranging from 20% [310] to 100%, because of the marked decrease in LEDD after STN-DBS implant [305,316,332].

Due to the relatively higher frequency and severity of Levodopa-induced dyskinesias and dystonic features in PARK2-related PD than PD noncarriers, GPi-DBS could be a valid target in this population. To date, four homozygous and compound heterozygous patients treated with GPi-DBS have been reported [310,335]. The rate of motor improvement was low (21%) at 12 months of follow-up [268,289,310], whereas Johansen and colleagues described a satisfactory autonomy 7 years after surgery in a PRKN carrier treated with unilateral GPi-DBS [335]. UPDRS IV score may improve up to 70%, mostly due to reduced dyskinesias despite the increase in dopaminergic medications needed for motor symptoms’ control [289,310]. 

Among non-motor symptoms, cognitive outcomes after DBS revealed no cognitive variations at 12–36 months of follow-up in several studies [331,333,335]. However, Lohmann and colleagues reported mild but significant deterioration in the Mattis Dementia Rating Scale in PARK2 carriers compared to patients with no mutations [330], and among these studies, only one heterozygous patient developed mania and hypersexuality after surgery [306]. 

To our knowledge, no specific studies, apart from single case reports or case series [287,336,337] are currently reported in the literature about Parkin PD carriers and other advanced therapies such as LCIG, radiofrequency, MR-guided focused ultrasound, gamma knife, and CASI. In more detail, Bohlega et al. reported good motor and non-motor outcomes (88% improvement in UPDRS-III, 74.4% improvement in Non-Motor Symptoms Scale, and 79.3% improvement in PDQ-39) in a p.T240M heterozygous PRKN mutation carriers with long (83 months) follow-up [336], while Foltynie et al. describe two PRKN patients treated with LCIG, but the outcomes were not specified [287]. With respect to CASI, Khan et al. described two patients carrying compound heterozygous PRKN mutations treated with CASI [337]. In one case, the outcome was not specified, whereas the second case displayed a motor benefit [337]. Nevertheless, both patients presented with severe psychiatric features before CASI, which improved after stopping the apomorphine infusion [337]. 

In conclusion, DBS-STN and DBS-GPi are the best advanced therapies in PARK2 carriers. Although controversial results have been reported in the literature, globally, PD patients with DBS, compared with those on oral treatment, have better control of motor symptoms, LEDD modification reduces dopaminergic fluctuations, there are no significant side effects on mood and cognition, and overall quality of life is improved. LCIG might be a good option for controlling levodopa-induced dyskinesia while subcutaneous injection of apomorphine could be a rescue strategy for improving sudden off-motor symptoms.

## 9. Conclusions

PD pathophysiology is associated with several processes, including α-synuclein aggregation, neuroinflammation, and mitochondrial and lysosomal dysfunction. Genetic mutation in specific portions of these pathways might respond to specific disease-modifying strategies through objective targeting of abnormal pathways. Research on these topics might help to predict which patients might be drug responders and non-responders, and potentially to a revolutionary approach to the disease progression. Currently, some genetic forms manifest better responses or worse side effects to a specific therapy, and the same therapies might modify the disease progression in certain mutation carriers. 

We strongly believe that the future of personalized therapy for PD will be mainly driven by precision genetic-driven approaches.

## Figures and Tables

**Figure 1 brainsci-12-01308-f001:**
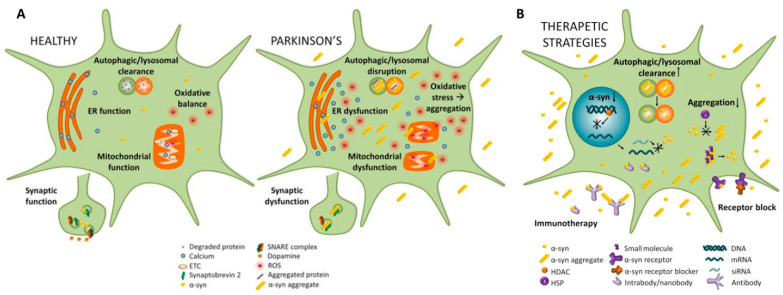
Implicated pathways for α-syn toxicity. (**A**). Cellular pathways in healthy subjects and PD patients are shown. In PD, there is a disfunction of clearance mechanisms based on autophagia and lysosomal activity leading to an accumulation of aggregated protein, which impair these homeostatic mechanisms promoting protein aggregation. In PD, endoplasmic reticulum and mitochondrial functions are impaired. In turn, pathological α-syn aggregates interact with several cellular proteins and functions, causing synaptic dysfunction and neurodegeneration. (**B**). Cellular pathways potential targets of disease-modifying therapeutic strategies. [35].

**Figure 2 brainsci-12-01308-f002:**
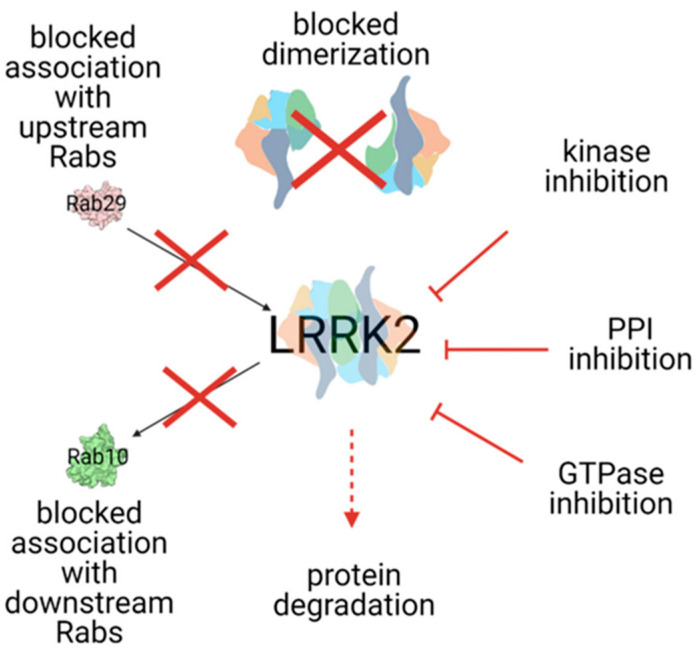
Summarized LRRK2 targeting strategies. LRRK2 dysfunction induces an increase in kinase activity, influencing PD development. Inhibition of LRRK2 kinase activity could prevente or restore several impaired cellular processes (see the text for details). Cross: blocked pathwas; full arrow: promoting activities; dashed arrow: final effect of the strategies. From [145].

**Table 1 brainsci-12-01308-t001:** Potential disease-modifying therapeutic strategies targeting α-Synuclein studied in synucleinopathies so far.

α-Syn Synthesis Inhibitors	Inhibiting α-Syn Aggregation	Inhibiting α-Syn Uptake	Immunotherapy	Autophagy-Enhancers
			**Passive**	
Antisense Oligonucleotides (ASOs) [55]	Intrabodies [56,57]	HSPGs binding [58]	Cinpanemab [59,60,61]	Rapamycin [62,63,64]
Amido-bridged nucleic acid-modified ASO [65]	Small molecules in vitro [66,67,68,69,70,71,72,73]	LAG3 [74]	Prasinezumab [75]	Nilotinib [76,77,78,79,80,81,82,83]
Exosome-ASO [84]	Small molecules in vivo	Neurexin 1β [85]	MEDI1341 [86]	Imatinib, Bafetinib, Radotinib [83,87]
Short interfering RNAs (siRNAs) [88,89]	NPT200-11[90,91]	APLP1 [85]	ABBV-080/mAb47 [92]	Vodobatinib [93,87]
2-Adrenergic Receptor agonists [84,94,95,96]	NPT088 [97]		Lu AF82422 [98]	IkT-148009/TKIs [87]
Small molecules	Anle138b [99]		**Active**	MPC inhibitors [100,101,102,103,104]
(a) Histone acetylase modulators [94,105]	LMTM [106]		PD01A [107,108]	
(b) IRE-block (Synucleozid) [109]	CLR01 [110]		PD03A [111]	
	KYP-2047 [112,113,114,115]		UCB-312 [116]	
	NPT100-18A [117]		ATV:aSyn [93]	
	cyclized-NDGA [118]			
	Fasudil [119]			
	Squalamine [120]			

## Data Availability

Not applicable.

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
