# Peer review of "Precision Medicine in Parkinson’s Disease: From Genetic Risk Signals to Personalized Therapy"

_brainsci, 2022, doi:10.3390/brainsci12101308_

Round 1
Reviewer 1 Report
This is a very comprehensive and insightful summary of the current state of science around genetically informed medicine in PD. The manuscript reads suprisingly well given the complexity and broadness of the topic.
I have only minor issues or rather suggestions: as the authors actually point out in the introduction, it is very difficult to infer causality or the place in a causal chain by a genetic signal. I really like that the authors start rather cautiously with the expression "risk signal". This is very appropriate because clearly, the risk of having PD is modified by some genetic variables. However, this does not necessarily mean that the modification is a primary part of the causal chain of pathology ("primum movens", as the authors call it). As an example, a gene cluster on chromosome 3 is a risk locus for severe COVID-19. However, we do not deduct that a genetic mutation on chromosome 3 causes COVID, because we know that a virus the culprit. As mentioned, the authors are clearly aware of this and point this problem out in some parts of the manuscript. In other parts, however, they are less cautious. For example, in section 6, page 17, they say that the alpha-syn phenomenon in sporadic PD may result from compensatory of protective cellular actions. Then again, they claim that "the role of Alpha-syn in triggering PD pathology is almost clear in specific genetic forms", such as SNCA. While that may be so, they may also consider that these mutations may anihilate the compensatory ot protective action that alpha-syn could convey. Along the same line, I suggest to change "provides a pathophysiological fingerprint" in "could provide". at the beginning of section 6, page 16.
Section 6: the part on mouse models is harder to understand than the rest of the manuscript. Please go over this part again and try to lay your arguments out more clearly.
Section 8 title : since the rest of the manuscript is about clinical management (of the future), I would suggest to say "current clinical management".
Section 8.3., page 20: non-motor: some repetitions occur, e.g., for autonomic symptoms. I find the section generally a bit long.
As a last suggestions (really only a suggestion): Since I like your term "risk signal" so much, I am tempted to suggest a slightly different title:
Precision medicine in Parkinson's Disease: from genetic risk signals to personalized therapy
Again, should you like it better yourself.
Unrelated to the content, I would like to suggest to add to the "Acklowledgements" part any sources of support from medical or pharmaceutical companies received by the authors in recent years.
Reviewer 2 Report
1. I suggest a more detailed discussion about the a-Syn reduction (could an excessive reduction of the "physiologic" a-Syn function be harmful?) Line:184-189
2. Rewording is required. Line: 509-510
3. I suggest a more detailed discussion about the magnitude of clinical benefit of different therapeutical options (both conventional and device aided) in different gene mutation
